# Study of Aroma Characteristics and Establishment of Flavor Molecular Labels in Fermented Milks from Different Fermentation Strains

**DOI:** 10.3390/foods14132237

**Published:** 2025-06-25

**Authors:** Rui Xu, Long Tang, Xing Gao, Xiaomo Han, Chen Liu, Huanlu Song

**Affiliations:** Laboratory of Molecular Sensory Science, School of Food and Health, Beijing Technology and Business University, Beijing 100048, China; xurui201015@163.com (R.X.); mtanglong0820@outlook.com (L.T.); gaoxing.ing@outlook.com (X.G.); tt529392945@163.com (X.H.); liuchen_lcc@163.com (C.L.)

**Keywords:** fermented milk, fermentation strains, aroma analysis, DHS, GC-O-MS, OAV

## Abstract

The aroma of fermented milk products is a key determinant of consumer preference. This study investigates the impact of different lactic acid strains on the aroma characteristics of fermented milk, identifies key volatile compounds, and establishes odor molecule labels to guide strain selection and modification. Sensory evaluation, dynamic headspace sampling (DHS), and gas chromatography olfactometry–mass spectrometry (GC-O-MS) were used to analyze 23 milk samples prepared with various lactic acid bacteria strains. A total of 74 volatile compounds were identified by GC-O-MS. Fermented milk P4 had the highest total volatile compound content (1566.50 ng/g). Flavor profiles were found to depend on strain metabolism rather than specific genera, with fermentation flavor quality enhanced by strains containing 2,3-butanedione, acetic acid, and sulfur compounds. Four distinct flavor clusters were established through molecular labels. These results provide targeted guidance for industrial strain selection and modification in fermented milk production, enhancing sensory appeal and consumer acceptance.

## 1. Introduction

Fermented milk is considered one of the largest classes of dairy products derived from animal milk, such as cow’s milk, through the lactobacillus fermentation process. The organoleptic properties of fermented milk—particularly its aroma, taste, and texture—are critical determinants of consumer choice in the marketplace. Fermented milk is highly favored by consumers due to its distinct flavor, which serves as a primary selection criterion among competing products. The unique aroma compounds of fermented milk consist of a combination of volatile compounds present in the raw milk and those produced during the fermentation of the raw milk [1]. This sensory-driven consumer preference underscores the commercial importance of optimizing flavor profiles, especially since over 100 volatile compounds have been identified in lactic acid bacteria-dominated fermented milks (*Streptococcus thermophilus* and *Lactobacillus bulgaricus*) [2]. The production of these volatile compounds is typically influenced by several factors, including the type and natural composition of raw milk, the type and proportion of lactic acid bacteria, fermentation time, and processing conditions [3].

Most odor compounds in fermented milk are present in very low or trace concentrations (μg/kg to mg/kg), yet they can still influence sensory perception. This might be attributed to the fact that organoleptic properties heavily depend on the relative balance of fat, protein, or carbohydrate-derived flavor compounds in a dairy product [1]. However, not all identified volatile compounds serve as essential characterizing compounds in fermented milks. Several studies have suggested that the production of acetaldehyde and 2,3-butanedione can be an important indicator for screening aroma-producing strains [2,4]. Additionally, studying the correlation between volatile compounds and flavor characteristics can provide insights into producing high-quality fermented milks [5].

During fermentation, different fermentation strains lead to the production of different volatile compounds, which can result in variations in the flavor of fermented milks. Some may produce unpleasant odors, such as fishy (dimethyl sulfide), sulfurous (dimethyl disulfide), and rancid (short-chain fatty acids like butanoic and hexanoic acids) in fermented milk products, thereby influencing the consumers’ preferences [6,7]. These preferences are shaped not only by the intrinsic aroma profile but also by cognitive factors such as evoked emotions, cultural associations, and past experiences with similar products. Tian et al. [8,9] screened aroma-producing lactic acid bacteria and investigated their application in improving the aromatic profile of yogurt. They found that using lactobacilli or probiotics as a supplemental culture for the co-fermentation of four probiotic strains in a traditional fermenter could increase the number of flavor compounds in yogurt, potentially leading to the development of a unique flavor profile. Different volatile compounds have different degrees of influence on the overall aroma profile of fermented milk. Currently, many dairy plants use the same fermentation strains, resulting in similarities or homogenization of fermented milk flavors, which is a key factor affecting the growth of the fermented milk industry [2].

With the advancements in molecular sensory science, the characterization of flavor based on sensory orientation has become increasingly rigorous. Gas chromatography–olfactometry combines instrumental analysis with human perception to achieve a more comprehensive analysis of food flavor [10]. While sensory evaluation captures the holistic human perception, including hedonic aspects, GC-O provides objective identification and characterization of individual odor-active compounds responsible for specific sensory attributes, overcoming limitations of subjectivity in pure sensory panels. However, GC-O analysis separates compounds from the food matrix, potentially altering perceived intensity or masking effects present in the whole product. Thus, the techniques are complementary: GC-O identifies key drivers, while sensory evaluation validates their impact in the complete food context. Food flavor labels can detail the flavor characteristics effectively. For instance, Chinese Baijiu can be categorized into several types based on the aroma characteristics, including strong-aroma-type Baijiu, soy sauce-aroma-type Baijiu, light-aroma-type Baijiu, etc. [11]. Liu et al. [2] classified fermented milks into milky-type, cheesy-type, fermented-type, and miscellaneous-type based on the sensory analysis results. Notably, compounds such as dimethyl sulfide (fishy aroma), dimethyl disulfide (sulfur off-flavor), and short-chain fatty acids (e.g., butanoic and hexanoic acids, associated with rancid notes) are key contributors to spoilage and undesirable aromas in fermented milk products. The clarification of the aroma characteristics produced by different strains after fermentation and the establishment of flavor molecular labels can further enhance the development and application of lactic acid bacteria strains in fermented milk. This type of flavor labeling not only facilitates factory production but also aids consumer preferences.

The aim of this study was to (1) determine the flavor compounds in 23 samples; (2) compare the differences between the strains and clarify their aroma characteristics; (3) conduct a systematic flavor evaluation of fermented milk flavors through molecular sensory science; (4) categorize aroma types of fermented milks based on sensory analysis and statistical analysis.

## 2. Materials and Methods

### 2.1. Materials and Reagents

Sodium chloride was purchased from Sinopharm Chemical Reagent Co. (Shanghai, China). Hexane (99%) was purchased from Fisher Chemical Company, Waltham, MA, USA; normal series alkanes (C7-C30) and 2-methyl-3-heptanone (99%) were purchased from Sigma-Aldrich, St. Louis, MO, USA; helium (99.9990%), nitrogen (99.9992%), and liquid nitrogen (99.9990%) were purchased from Beijing Helipu Beifen Gas Industry Co., Beijing, China.

### 2.2. Preparation of Fermented Milk Samples

Lactic acid bacteria strains and ultra-high-temperature (UHT) sterilized bovine milk were provided by Inner Mongolia Yili Industrial Group Co., Ltd. (Hohhot, Inner Mongolia, China). Strain information is presented in Table 1. The letter A denotes the *Lactobacillus acidophilus*, L refers to *Lactobacillus delbrueckii* subsp. *bulgaricus*, P stands for *Lactobacillus casei*, and S corresponds to the *Streptococcus thermophilus*. Due to confidentiality requirements, the specific strain types used in fermentation are represented by alphanumeric codes (e.g., L3, P4, and S9) to protect proprietary information. Raw milk was replaced with the letter N. Sterilized at 85 °C for 30 min (Model ZW-A, Zhenjiang Instrument Co., Ltd., Zhenjiang, China), the strain was inoculated into the sterilized raw milk at 1% and incubated at 42 °C for 6 h. Then, it was placed in a refrigerator at 4 °C for 24 h to complete the fermentation.

### 2.3. Sensory Evaluation

The samples were subjected to a Quantitative Descriptive Analysis (QDA). Sensory evaluation was conducted by 12 sensory experts (6 males and 6 females, aged between 22 and 30) recruited from the Laboratory of Molecular Sensory Science, Beijing Technology and Business University Beijing, China, with an average age of 24 years. Each member of the sensory evaluation staff was trained according to ISO 8589–2007 [12] and had more than one year of sensory evaluation experience. A total of 30 g of the sample was accurately weighed and placed in an 80 mL wide-mouth headspace flask. Panelists did not know any information about the samples before the sensory evaluation. First, all samples were provided to the sensory panelists for evaluation and analyzed within 1 h. The panelists were asked to describe the odor composition of the samples and complete the collection and screening of odor attributes. Each sample was assessed thrice by every panel member. Through group discussion and the literature review, eight sensory descriptors were identified for the samples. The system of flavor descriptors for fermented milks included fermented (butanoic acid), milky (full-fat pasteurized milk), sweet (2-heptanone), fishy (raw milk), sour (acetic acid), green-like (green apple), sulfur (hard-boiled, mashed egg), and buttery (fresh, unsalted butter) aromas for a total of eight flavor descriptors. The overall flavor perception of the sample was expressed as “preference”. Then, the panelists were asked to evaluate each sample at 5-min intervals between adjacent samples and sniff coffee beans to relieve olfactory fatigue. A 5-point scale was used to rate the perception: 0 points for no perception; 1 point for just perceptible; 2 points for weak perception; 3 points for moderately strong perception; 4 points for strong perception; and 5 points for very strong perception [13]. The samples to be tested were assessed blindly and randomly numbered with a random three-digit number. The sensory room was maintained at a constant temperature of about 21 °C and the humidity was maintained at about 65% with fresh and odorless air. Each sample was tested in triplicate and the mean value was obtained.

### 2.4. Dynamic Headspace Sampling (DHS)

A total of 90 g of fermented milk and 27 g of sodium chloride, accurately weighed, were placed in a conical flask and mixed thoroughly. Afterwards, 100 g of the mixed sample solution was transferred to a 250 mL three-necked dynamic headspace flask, and 10 μL of 2-methyl-3-heptanone (0.816 μg/μL, dissolved in n-hexane) was added as the internal standard reference. Then, the top cap and the caps on both sides of the dynamic headspace bottle were tightened immediately. The thermostatic circulating water bath was set at 40 °C, and the magnetic stirrer was adjusted to 800 rpm. The system was stabilized and equilibrated under these conditions for 20 min. Later, nitrogen (purity 99.9992%) was passed into one end of the outlet on both sides of the bottle, respectively, and one end was inserted into the adsorption column containing Tenax adsorbent material. Nitrogen was used to purge the system at a flow rate of 100 mL/min for 45 min, and the temperature of the system was maintained at 40 °C [14,15]. Following the complete adsorption of the volatile compounds in the sample onto the Tenax TA thermal desorption tubes(Gerstel GmbH & Co. KG, Mülheim an der Ruhr, Germany), the water vapor remaining on the glass tube walls was removed using nitrogen at a mild flow rate. Finally, the de-watered Tenax thermal desorption tubes were placed into a thermal desorption system (TDU) for thermal desorption and separated and analyzed using a Thermo Scientific TriPlus RSH autosampler-coupled thermal desorption unit (TDU) (Waltham, MA, USA), followed by GC-O-MS. Each sample was tested in triplicate. Among them, the TDU warming-up procedure was as follows: the starting temperature was 50 °C, which was maintained for 1 min and then increased to 280 °C by 60 °C/min and held for 10 min. The CIS warming procedure was as follows: firstly, the temperature of CIS was rapidly lowered to −50 °C by ultra-low-temperature liquid nitrogen, and after the desorption of TDS was completed, the system was warmed up to 280 °C at 10 °C/min and held for 5 min.

### 2.5. Gas Chromatography–Olfaction–Mass Spectrometry (GC-O-MS)

GC-O-MS analysis was performed using an Agilent 7890B gas chromatograph (Agilent Technology Inc., Santa Clara, CA, USA) coupled with a 5977B mass spectrometer (Agilent Technology Inc., Santa Clara, CA, USA) and an Olfactory Detection Port (ODP4, Gerstel GmbH, Mülheim an der Ruhr, Germany). The parameters of gas chromatography conditions were as follows: initial temperature 35 °C, held for 3 min, then warmed up to 80 °C at 3 °C/min, held for 1 min, then warmed up to 230 °C at 10 °C/min, held for 5 min, and then ran at 250 °C. Separations were carried out using DB-WAX (30 m × 0.25 mm, 0.25 μm film thickness; J&W Scientific, Folsom, CA, USA),and DB-5 columns (30 m × 0.25 mm, 0.25 μm film thickness; J&W Scientific) with different polarities to compare the separation results and corroborate each other. The mass spectra were obtained under the following conditions: an electron impact (EI) ion source with an electron energy of 70 eV; a transfer line temperature of 280 °C; an ion source temperature of 230 °C; a quadrupole temperature of 150 °C; a solvent delay of 5 min; and a mass scan range of 35–350 *m*/*z*. The ion source temperature was set at 230 °C, the solvent delay was set at 150 °C, and the solvent delay was set at 5 min. Olfactory detector (Sniffer 9100; Brechbuhler, Schlieren, Switzerland): interface temperature of 200 °C. Moist nitrogen gas was introduced during the sniffing test to prevent drying out the panelists’ nasal passages, which may affect the results.

### 2.6. Characterization and Quantification of Odor Compounds

All odor compounds were characterized by combining the results of the NIST17 library, retention index (RI), and odor property (O). Semi-quantitative determination was performed using an internal standard method, in which the concentration of each odor compound was calculated as follows:(1)$$C_{i}=C_{is}\;*\;A_{i}/A_{is}$$
where *C_i_* represents the concentration of the target compound in μg/μL; *C_is_* represents the concentration of the internal standard compound in μg/μL; *A_i_* represents the peak area of the target compound; and *A_is_* represents the peak area of the internal standard compound [16].

### 2.7. Data Processing and Analysis

Data analysis used IBM SPSS 25 and SIMCA 14.1. Sensory data underwent three-way ANOVA (Sample × Panelist × Replicate) with Tukey’s HSD post hoc test (*p* ≤ 0.05). Spearman’s rank correlation analyzed relationships between attributes/compounds. Cluster analysis (Ward’s method, Euclidean distance) was applied to sensory scores and relative OAVs. Partial least squares regression (PLS-R) modeled sensory attributes (X) against volatile compounds (Y), with model validity confirmed by permutation testing (*n* = 100). Confidence ellipses (95%) were generated for PLS-R score plots.

### 2.8. Experimental Design Flowchart

Figures should not be inserted right under the title. Please add a short of paragraph of the section here and mention the Figure.

Figure 1 is the overall design of the experimental process.

## 3. Results and Discussion

### 3.1. Sensory evaluation analysis

The fermented strains that ranked first in different sensory attributes are listed in Table 2. Among all sensory attributes, different strains showed differences in the performance of different sensory attributes.

The correlation analysis of the preference scores with the sensory attributes concluded strong positive correlations between fermented and sour, sweet and sour, and fermented and milky aromas; in addition, it found negative correlations between fishy and sulfur, green-like and sulfur, and green-like and buttery aromas (Figure 2). The level of buttery, milky, sour, and sweet aromas had a significant positive effect on the preference level, and some degree of green-like and fermented aromas also had an effect on the preference level. The correlation was indicated by the colors in the figure, red represents a strong correlation, blue represents a weak correlation. Notably, the strong correlation between buttery aroma and 2,3-butanedione (Figure 3b) corroborates its role as a universal marker for dairy flavor [15].

The sensory evaluation results were characterized using a clustered heatmap. A clustered heatmap (Figure 3) was constructed using Ward’s linkage method and the Euclidean distance metric. Rows represent samples, columns represent the eight sensory attributes, and color intensity (green to red) corresponds to the normalized sensory score magnitude. As shown in Figure 3a, the top dendrogram indicates clustering based on sensory attributes. Cluster analysis separates samples into three groups: Group 1 (N, L3, and P3) retains raw milk characteristics; Group 2 (P4 and S9) exhibits high buttery/milky intensity; Group 3 (others) shows dominant fermented/sour notes. In each sensory attribute and rating, preference, buttery, milky, and sweet aromas were generally rated higher and clustered into one category among all samples, while sour, fermented, and green-like aromas were clustered into one category. Fishy and sulfur aromas were mostly low in all fermented samples and clustered into one category but those were the typical negative flavors in fermented milks. As shown in Figure 3b, the dendrogram on the left indicates sample clustering based on sensory attributes. The clustering between the samples indicated that the pre-fermentation samples had a strong fishy aroma, and after fermentation, L3 and P3 shared similarities with sample N compared to the other fermented samples. This is probably because these three retained the buttery, milky, and sweet notes of the unfermented samples along with increased fermented and sour aromas after fermentation. Additionally, the other samples differed significantly from these three samples while clustering in another large grouping. As shown in Figure 3, the fermented samples had many sensory differences compared to sample N.

### 3.2. Qualitative and Quantitative Results of Volatile Compounds in Fermented Milks

A total of 23 raw and fermented milk samples were analyzed for qualitative and quantitative analysis of volatile compounds using an internal standard semi-quantitative method suitable for the analysis of large sample sizes. The results are detailed in Appendix A. A total of 74 volatile compounds were detected, including 23 odor-active compounds. Additionally, the common key odor-active compounds in fermented milks were detected. The sum of the concentrations of each class of compounds in 22 fermented milk samples (A2–S11) and raw milk samples (N) was determined using stacked bar charts to visualize the differences in the content of different compound classes in different samples. The results were obtained using stacked bar charts, as summarized in Figure 4. The results indicated that different strains before and after fermentation had a great influence on the flavor of fermented milk and affected it to different degrees. The 74 volatile compounds included 7 esters, 17 alcohols, 8 aldehydes, 13 acids, 20 ketones, 4 sulfur-containing compounds, and 5 other compounds. Among them, seven compounds, 2(*5H*)-furanone (buttery), 3-methylbutanal (peachy), ethyl acetate (fruity), isopropyl alcohol (moldy), butanol (whisky), methyl heptenone (citrus), and 4-methylphenol (green-like) were only detected in sample N. This result indicated that after fermentation, lactic acid bacteria consumed or decomposed these odor compounds by metabolism and converted them into other compounds, which in turn changed the flavor of the raw milk samples and contributed to the characteristic flavor of fermented milk. Among them, L3, P4, and P7 produced more flavor substances after fermentation. Alcohols were most abundant in the raw milk samples and fermented samples P4, A2, and S9; aldehydes were most abundant in N, S8, and S2; acids were most abundant in P4, L3, and P7; ketones were most abundant in P4, P7, P5, A2, and P3; ester compounds were most abundant in N and L1; sulfur compounds were most abundant in S9, N, and A2, while S9 generates elevated sulfur compounds (9.15 ng/g dimethyl sulfone). P4 and P7 had the highest total volatile compounds of 1566.50 ng/g and 1027.47 ng/g, respectively. In contrast, P1, L4, and P2 had the lowest total compound contents of 286.39 ng/g, 231.14 ng/g, and 135.39 ng/g, respectively.

Alcohols were also a key class of volatile compounds in fermented milks. Alcohol compounds were the most abundant in raw milk samples but the thresholds for alcohol compounds were often high and did not contribute much to the aroma. In contrast, the content of alcohol compounds in fermented milk was lower than in raw milk samples. It is reported that the presence of most alcohols serves as an intermediate that contributes to the formation of esters and reduces the harsh sour aroma caused by acid compounds, making the fermented milk flavor milder and more pleasant [17]. 2,3-butanediol, a reduction product of 3-hydroxy-2-butanone, is a major contributor to the creamy flavor [18]. However, it could not be smelled in this study due to its high threshold and a weak odor-activity.

The primary ketones identified in this study included 2,3-butanedione, 3-hydroxy-2-butanone, 2,3-pentanedione, acetone, 2-heptanone, 2-pentanone, and 2-nonanone, which was consistent with the previous studies [6,15]. Although acetone and 2-butanone are two common volatile compounds in fermented milks, 2-butanone was not smelled in this study. Cárcoba et al. [19] also found that 2-butanone had little effect on the flavor of dairy products.

Butyl acrylate was the first odor-active compound identified in fermented milks, which is often used as a chemical raw material and widely employed in plastics, food packaging, and other fields [20,21]. However, Straathof et al. [22] found that acrylic acid, the key precursor of butyl acrylate, can be synthesized by microbial fermentation and can be esterified with the original butanol in fermented milks to produce butyl acrylate with a tropical fruit aroma.

Acetic, propionic, butanoic, valeric, hexanoic, and octanoic acids detected in this study have also been previously identified by [23] in co-cultures of *Streptococcus thermophilus* and *Lactobacillus bulgaricus*. Some studies have also hypothesized that C2–C4 acids are usually metabolized by lactic acid bacteria, whereas C4–C20 acids are mainly formed by lipolysis [24]. Of the 15 acids identified, only 3 were odor-active, namely acetic, butanoic, and hexanoic acids. Butanoic and hexanoic acids are also the key odor-active compounds of fermented dairy products such as Swiss cheese [25] and Turkish yogurt Ayran [26]. Acetic acid is the most concentrated compound and has a vinegary aroma profile. At high concentrations, it can have an irritating negative impact on the overall flavor profile. However, the presence of other odor compounds diluted the effect of these stimulating compounds, such as octanoic acid, which had a rancid odor. This odor was weakened in its odor profile or was even inactive in aroma, making them undetectable to the smeller [17].

Odor compounds of the same species have similar structural characteristics; however, their aroma profiles cannot be simply differentiated exclusively according to the class of compounds. Therefore, the content profiles of the key odor compounds in different strains were summarized, and the odor compounds were ranked based on their odor activity and importance in fermented milk, as shown in Table 3. The importance of the odor compounds was categorized into four levels: 1. Atypical odor compounds in fermented milk that were not odor-active. 2. Typical odor compounds in fermented milk that were not odor-active; 3. Atypical odor compounds in fermented milk that were odor-active. 4. Typical odor compounds in fermented milks that were odor-active. In this study, hexanal, 2,3-butanedione, 2,3-pentanedione, 2-nonanone, 2-heptanone, methyl nonyl ketone, acetic acid, hexanoic acid, and butanoic acid were the key flavor compounds of fermented milks. Key compounds responsible for unwanted aromas include the following: butanoic acid (cheesy, rancid notes at high concentrations); hexanoic acid (sweaty off-flavor); dimethyl sulfide (fishy aroma); dimethyl trisulfide (sulfurous defect). The combined results of Figure 5 and Table 3 and Table 4 demonstrated that there were obvious metabolic differences among the different fermentation strains, which further led to the production of different flavor compounds. Moreover, the differences in some key flavor compounds directly affected the flavor of the fermented products.

### 3.3. Analysis of Similarities and Differences in Aroma Characteristics Among Fermented Strains

A total of twenty-two fermented milk samples were produced by twenty-two different strains belonging to two different genera, four different species, and twenty-two different types. The different fermentation agents will be discussed at different levels. A total of 13 key odor-active compounds were identified, including 2,3-butanedione, 2,3-pentanedione, acetic acid, butanoic acid, hexanoic acid, methyl nonyl ketone, 2-heptanone, 2-nonanone, nonanal, hexanal, heptanol, hexanol, and dimethyl sulfide. Five common odor compounds were detected in fermented milks, including acetone, 3-hydroxy-2-butanone, benzaldehyde, *γ*-decalactone, and isopentenol. The above 18 odor compounds included almost all typical odor-active compounds that constitute the basic aroma profile of fermented milks; however, there was no bias to produce it only in a particular genus. This finding suggests that the formation of the basic aroma profile of fermented milks does not depend on a single genus. However, some atypical fermented milk aroma compounds were also present in the samples from different genera, showing different aroma profiles for different genera—such as 3-methyl-2-butenal (sweet), (*E*)-2-octenal (cucumber), and furfural (bakery)—that were only detected in the Streptococcus genus; five compounds, dimethyl trisulfide (sulfur), dimethyl disulfide (onion), (+)-limonene (citrus), *γ*-octanolactone (creamy), and 2-pentanone (sweet) were detected only in *Lactobacillus* spp. The presence of these compounds resulted in subtle differences in the aroma composition of fermented milks of different genera. However, the combined results with sensory data elucidated that under current production processes and conditions, these compounds did not contribute to the overall aroma profile to the extent that they can be clearly perceived by the senses. The detection of these compounds revealed that there was metabolic variation between genera in the same substrate. The genus-level metabolic differences (e.g., *Streptococcus*-exclusive 3-methyl-2-butenal) explain flavor diversification beyond core compounds. This echoes Tian et al. who emphasized strain-level screening over genus-level generalization for aroma optimization [5].

There may be limitations in the generalization of aroma for the fermentation characteristics of different strains due to the uncertainty of the number of types contained in each strain and the differences in metabolic capacity. It should be noted that the aroma characteristics summarized below in terms of strains represent only the strain characteristics exhibited by the existing strains. The eleven key odor-active compounds included the following: 2,3-butanedione, 2,3-pentanedione, acetic acid, butanoic acid, hexanoic acid, methyl nonyl ketone, 2-heptanone, 2-nonanone, nonanal, hexanal, and hexanol. The five common odor compounds included the following: acetone, 3-hydroxy-2-butanone, benzaldehyde, *γ*-decalactone, and isopentenol. In A, L, P, and S, 12 key odor-active compounds and 5 common odor compounds were not biased to be produced only in one strain species. Similar to the analysis of the commonality of different genera, it was predicted that the formation of the basic aroma profile of fermented milks should not be dependent on a single strain species. However, unlike different genera, the aroma profiles already showed partial variability in the samples fermented by different species—such as dimethyl sulfide (cabbage) and heptanol (herbal)—were not detected in the samples fermented by strain A. 3-Methylbutanoic acid (sweaty), dimethyl trisulfide (sulfur), and *γ*-octanolactone (creamy) were detected only in samples L and P; butyl acrylate (fruity) was detected only in samples L and S; dimethyl disulfide (onion) was detected only in samples L and A; (+)-limonene (citrus) was detected only in samples A and P. Among the flavor compounds detected in the fermented milks of different types, only three compounds, 2-heptanone (fruity), 2-nonanone (sweet), and acetone were common in all types. These findings suggested that strain L was more capable of producing richer flavor substances than the other strains in single-strain fermentation. In contrast, strain A produced the least amount of flavor substances and strain S produced the middle amount of flavor substances. This result provided insights into mixed-strain fermentation. By selecting specific bacterial cultures, producers can tailor the volatile compound profile to enhance desirable notes (e.g., buttery and milky) and suppress spoilage-associated aromas (e.g., fishy and sulfurous). Strains such as P4 and L3 (Table 4), which produce high levels of 2,3-butanedione and acetic acid, yield flavor profiles most preferred by consumers (Figure 2).

### 3.4. Establishment of Flavor Molecular Labels for Different Fermented Strains

Following qualitative and quantitative analysis of the odor compounds from different fermenter strains, a preliminary screening of the compounds was carried out using relative OAV to further screen these odor compounds and analyze them statistically. The larger the relative OAV, the greater the contribution of the odor compound to the overall flavor of the sample. Therefore, the relative OAV of the 26 odor-active compounds shown in Figure 5 were screened and summarized for clustering heatmap analysis, as shown in Figure 5a. Red represents that the odor compound has a higher OAV.

The clustering results demonstrated that the differences in flavor substances between the samples before and after fermentation significantly differentiated the fermented milk from the raw milk samples. This phenomenon indicated that due to the metabolism of the strains in cow’s milk, the original compounds were consumed, transformed, or new compounds were produced, resulting in a change in flavor. These altered flavor substances enabled accurate strain characterization. *γ*-decalactone (creamy), furfural (bakery), 3-methylbutanoic acid (sweaty), furfuryl alcohol (toast), ethyl acetate (fruity), 3-methylbutanal (malty), 4-methylphenol (green-like), heptanol (herbal), hexanol (fruity), nonanal (floral), and 2-pentylfuran (soy milk) were the most abundant in unfermented milk samples. Although it is possible to distinguish unfermented samples from fermented samples directly from the sensory evaluation by virtue of the presence or absence of distinct sour and fermented aromas, these sensory attributes were not sufficient to distinguish the pre- and post-fermentation samples upon clustering the sensory attributes in a heatmap. This finding suggested that there was a certain degree of overlap in the overall aroma profiles of the fermented milks with those of the original milks. In contrast, when the relative OAV of some key odor compounds were analyzed statistically, the samples before and after fermentation were significantly differentiated in the clustered heatmap. This is probably because some different compounds can provide similar aroma characteristics. Secondly, the relative OAV of the compounds was correlated with the sensory evaluation results to further investigate the contribution of different compounds to different flavor profiles of fermented milks and the interrelationships between compounds and sensory attributes. As shown in Figure 5b, there was indeed a relationship between the compounds and different sensory attributes to varying degrees. The first correlation was between sensory attributes and compounds: the fermented aroma showed a very strong positive correlation with butanoic and hexanoic acids (*p* < 0.001) and a negative correlation with compounds such as furfural, 3-methylbutanal, and ethyl acetate. Milky aroma showed a very strong positive correlation with 2,3-pentanedione and methyl nonyl ketone (OAV > 100), a strong positive correlation with compounds such as 2-nonanone and 2-heptanone, and a negative correlation with trans-2-octenal. Fishy aroma showed a strong positive correlation with dimethyl sulfide (*p* < 0.01), while sour aroma showed a strong positive correlation with hexanoic acid and butanoic acid. Green-like aroma showed a strong positive correlation with nonanal. Sulfur aroma showed a strong positive correlation with dimethyl disulfide and dimethyl trisulfide and a negative correlation with nonanal. Buttery aroma showed a strong positive correlation with 2,3-butanedione. The second correlation was between compounds, such as some of the key odor-active compounds: 2,3-butanedione and acetic acid showed a strong positive correlation. The compounds showed a positive correlation with 2,3-butanedione, a typical odor compound of fermented milks, indicating that they might be the key products of fermentation. In contrast, there was a strong positive correlation between the compounds such as nonanal, furfural, *γ*-decanolactone, 2-pentylfuran, 3-methylbutanal, and furfuryl alcohol; most of these compounds were detected only in unfermented samples, indicating that most of them were metabolized or transformed after fermentation. While 2,3-pentanedione and butanoic acid showed a strong positive correlation, butyric acid and hexanoic acid showed a very strong positive correlation; these compounds also showed a strong correlation with fermentation.

After analyzing the contribution of odor compounds to the overall aroma and different sensory attributes, the fermented milk samples were differentiated from different strains using partial least squares regression analysis (PLS-R). PLS-R analysis showed the relationship between sensory attributes (X) and odor compounds (Y) of different fermentation samples, as shown in Figure 6. The contribution rates of X and Y for factor 1 and factor 2 were 34.9% and 17.7%. Some fermented milk sample scatters are more clustered in the loading diagram but the whole was also differentiated in terms of different compounds and sensory attributes, which can be mainly categorized into four groups. These four groups included the characteristic flavor Group 1 consisting of S9, A2, P4, and P7; Group 2 consisting of S10, S5, S2, L1, and P3; Group 3 consisting of S1, L4, S4, L2, S8, S11, and P5; and an Outlier Group 4 consisting of P2, P1, and P6. Group 1 was characterized by strong fermented aroma, and the key attribute that distinguished them from the other samples was buttery. The main characteristic compounds included 2,3-butanedione, acetic acid, and sulfur-containing compounds. Additionally, the four attributes of fermented, sweet, sour, and milky differed from the other groups, and their main characteristic compounds included 2,3-pentanedione, hexanoic acid, butanoic acid, and methyl nonyl ketone. In this study, these four were typical of fermented flavor. In all other groups, the post-fermentation flavor could not be perceived. Fermentation strains can be selected according to the flavor preferences of consumers. 2,3-butanedione, acetic acid, and sulfur compounds were the most influential variables driving Group 1 clustering (S9, A2, P4, and P7), aligning with their high buttery and fermented sensory scores. Conversely, Group 2 (S10, S5, and S2) was characterized by nonanal and hexanal, correlating with green-like and milky attributes. Based on the PLS loading diagram, the summarized flavor molecule labels are summarized in Table 4. The sensory attributes and compounds in the label of a flavor molecule did not refer to the most abundant or the most prominent of the compounds but rather to the most important indicators that differentiated them in a sample size of 22 fermenter strains. Overall, the results confirmed that 2,3-butanedione, 2,3-pentanedione, methyl nonyl ketone, acetic acid, hexanoic acid, butanoic acid, and sulfur-containing compounds significantly enhance the flavor of fermented milks.

## 4. Conclusions

In summary, the aroma compounds in 23 fermented milks were analyzed by sensory-directed flavor analysis. The results showed a strong positive correlation between preference and different sensory attributes such as buttery, milky, sour, and sweet without specific characterization of different genera or species of bacteria by specific sensory attributes. Among the flavor compounds detected in fermented milks of different types, only three compounds—2-heptanone (fruity), 2-nonanone (sweet), and acetone—were common in all types. Furthermore, although the production of the basic flavor profile of fermented milks was not dependent on a particular genus or species, the metabolic differences between different types resulted in the absence of some key aroma-active compounds that form the basic flavor profile of fermented milks, which can directly lead to changes in the flavor profile of fermented milks. Therefore, while selecting fermenting strains, it is important to choose strains with prominent organoleptic properties such as buttery, milky, sour, and sweet aromas, i.e., strains containing 2,3-butanedione, 2,3-pentanedione, methyl nonyl ketone, acetic acid, hexanoic acid, butanoic acid, and sulfur-containing compounds. The differences in flavor substances between the samples before and after fermentation significantly differentiated the fermented milk samples from the raw milk samples. These strains could be classified into four groups according to their relevant flavor profiles, enabling targeted strain selection (e.g., Group 1: P4/L3 for buttery notes; Group 2: S9 for a balanced profile) and their corresponding flavor molecular labels could be listed within the groups, which will provide guidance for further development and utilization of the strains such as mixed fermentation. This study establishes the first flavor molecular labeling system for fermented milk strains, enabling targeted selection of strains for desired sensory profiles. The PLS-R model and odorant clustering strategy provide a transferrable framework for precision fermentation in dairy industries.

## Figures and Tables

**Figure 1 foods-14-02237-f001:**
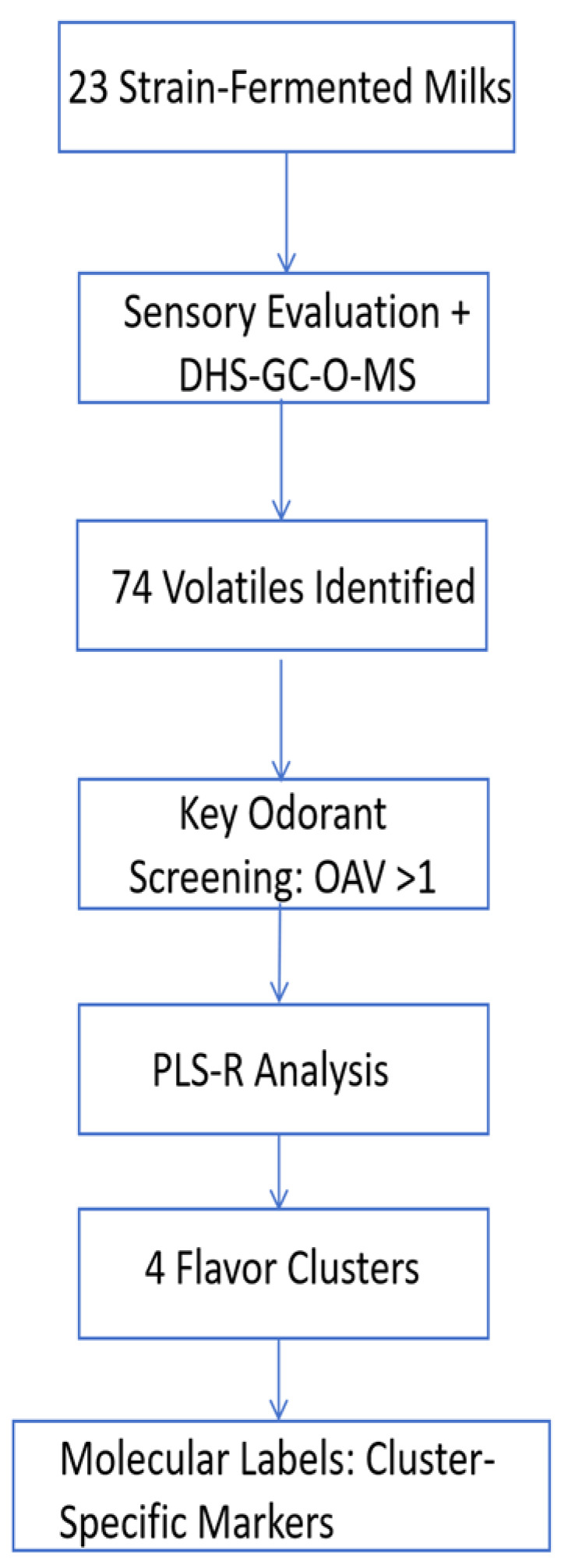
Experimental design flowchart.

**Figure 2 foods-14-02237-f002:**
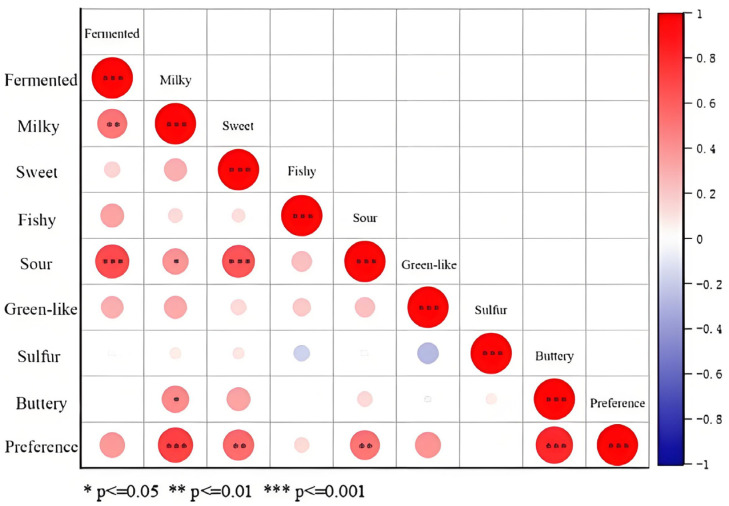
Correlation analysis diagram of sensory evaluation. (Employing Spearman’s correlation analysis: * *p* ≤ 0.05, significant correlation; ** *p* ≤ 0.01, strong significant correlation; *** *p* ≤ 0.001, extremely strong and significant correlation.)

**Figure 3 foods-14-02237-f003:**
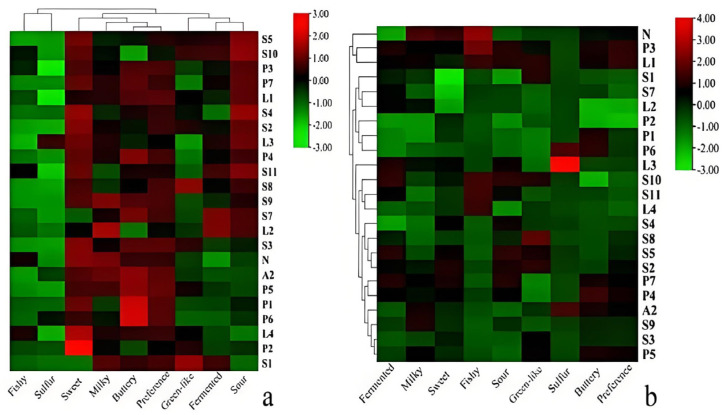
(**a**) Clustered heatmap of the ratings of each sensory attribute for the same sample. (**b**) Clustered heatmap of the ratings of each sensory attribute across all samples. The color scale of the heatmap represents the intensity of the senses, with red indicating strong senses and green indicating weak senses.

**Figure 4 foods-14-02237-f004:**
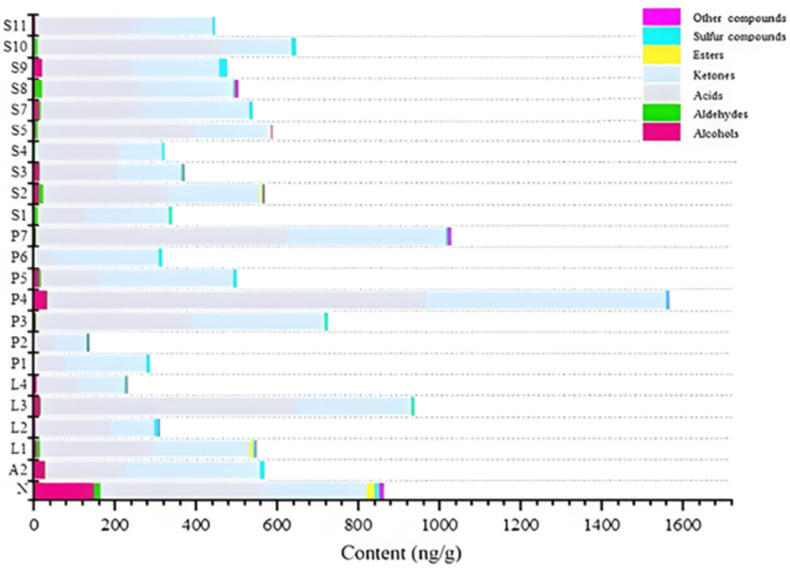
Stacking histogram of classified content of compounds. The color scale of the graph showed the comparison between the concentrations of different compounds.

**Figure 5 foods-14-02237-f005:**
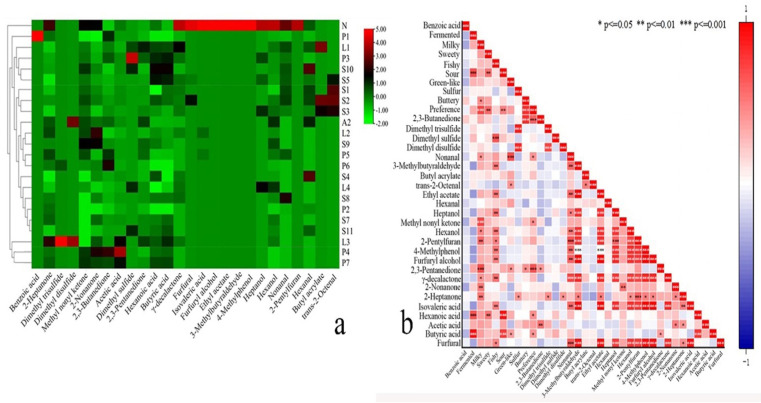
(**a**) Heatmap of the relative OAV of odor compounds. (**b**) Analysis of the correlation between odor compounds and sensory attributes. (Employing Spearman’s correlation analysis: * *p* ≤ 0.05, significant correlation; ** *p* ≤ 0.01, strong significant correlation; *** *p* ≤ 0.001, extremely strong and significant correlation).

**Figure 6 foods-14-02237-f006:**
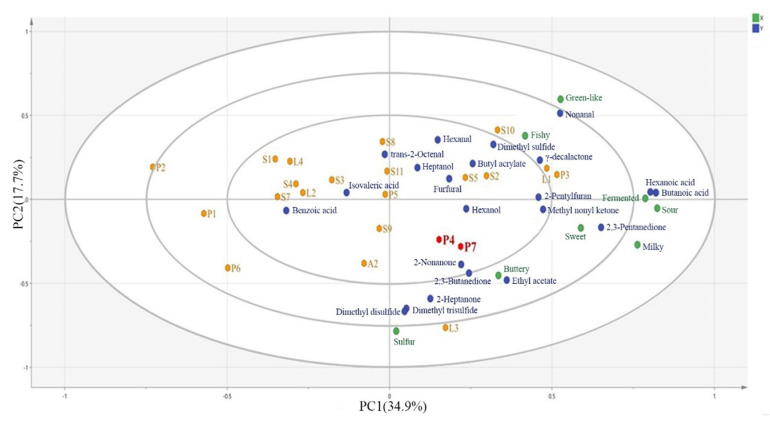
PLS-R analysis of different fermented milk samples. (Variables X and Y represent sensory attributes and different compounds.)

**Table 1 foods-14-02237-t001:** Fermentation strain information.

Species	A	L				P							S									
Type	A2	L1	L2	L3	L4	P1	P2	P3	P4	P5	P6	P7	S1	S2	S3	S4	S5	S7	S8	S9	S10	S11

**Table 2 foods-14-02237-t002:** Ranking of strains with sensory properties.

Sensory	Buttery	Milky	Sweet	Fishy	Fermented	Sour	Green-Like	Sulfur
Top ranked	P4	S9	A2	P3	S10	S5	S8	L3

**Table 3 foods-14-02237-t003:** Content characteristics of key odor compounds in different strains.

No.	Compounds	Odor	Strain Species	Importance (1–4)
1	2,3-Butanedione	Strong butter flavor	A\P	4
2	Acetic acid	Sour	P\L\A	4
3	2,3-Pentanedione	Sweet and buttery	A\L	4
4	Hexanoic acid	Sweaty and sour	S	4
5	2-Nonanone	Sweet	L\P	4
6	2-Heptanone	Fruity	L\A	4
7	Butanoic acid	Cheesy	S\P\L\A	4
8	Methyl nonyl ketone	Sweet	S	4
9	Hexanal	Fresh, grassy	S	4
10	Nonanal	Rosy	S\A	3
11	Hexanol	Fruity, fresh	A\L	3
12	Heptanol	Fresh	L\S	3
13	Dimethyl sulfide	Fishy, garlic	P	3
14	3-Hydroxy-2-butanone (acetoin)	Sweet, creamy	P	2

**Table 4 foods-14-02237-t004:** Flavor molecular tags of different strains.

Group	Type	Flavor Molecule Label
1	S9, A2, P4, P7, L3	2-Nonanone, 2-Heptanone, 2,3-Butanedione, Acetic acid, Dimethyl disulfide, Dimethyl trisulfide
2	S10, S5, S2, L1, P3	Nonanal, *γ*-Decalactone, 2-Pentylfuran, Hexanol, Furfural, Heptanol, Butyl acrylate, Dimethyl sulfide, Hexanal
3	S1, L4, S3, S4, L2, S8, S11, P5, S7	Trans-2-Octenal, Isovaleric acid, Benzoic acid
4	P2, P1, P6	(Outlier group)

## Data Availability

The original contributions presented in this study are included in the article/Appendix A. Further inquiries can be directed to the corresponding author.

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
