# Peer review of "Study of Aroma Characteristics and Establishment of Flavor Molecular Labels in Fermented Milks from Different Fermentation Strains"

_foods, 2025, doi:10.3390/foods14132237_

Round 1
Reviewer 1 Report
Comments and Suggestions for Authors
The manuscript explores the characterization of the aroma profile in fermented milk using different lactic acid bacteria strains. The use of GC-O-MS and sensory analysis, in combination, add depth to the methodological design, and the attempt to create molecular flavor labels is both original and valuable for product development. A few improvements in clarity, formatting, and structure would enhance the manuscript’s readability and scientific rigor.
-Clarify whether the aroma compounds are predictive, discriminatory, or descriptive;
-Strengthen the connection between specific volatile compounds and sensory attributes;
-Indicate which compounds were most influential in the PLS model and how results relate to sensory clustering;
-Briefly explain why the five bacterial strains were chosen (e.g., industrial relevance, metabolic diversity);
-Line 3 (Title): Suggestion: change "fermentation strains" for "lactic acid strains";
-Line 8: Change "authors. Email address" to "Author’s email address";
-Line 11 (Abstract): Suggestion: change "microbial strains" for "lactic acid strains";
-Line 99: “experience..” Remove the extra period.
-Figures 2, 3, 4, and 5: Improve axis label font size and quality to enhance readability;
-Table 5: Clarify what makes these compounds “labels”—are they predictive, discriminatory, or just characteristic?
-Decimal places are not consistently formatted across tables.
Author Response
It is attached here

Reviewer 2 Report
Comments and Suggestions for Authors
Dear Authors,
This manuscript is cover topic related to the aroma study of fermented milk products obtained after application of different fermentation strains.
In the abstract are missing values of the most important results. Insert it.
Highlight in the introduction that organoleptic properties of food such as fermented milk products are key for consumer to choose specific product.
Highlight in the introduction which compounds are responsible for spoilage and unwanted aromas of fermented milk products.
In the subsection 2.2 are missing information about milk.
What kind of milk was used?
Insert city, province of company which provided milk.
In the subsection 2.4. is missing information regarding the type of system which was used for analysis.
Insert type, name of producer, city and country.
In the subsection 2.5. is missing information regarding the type of system which was used for analysis.
Insert type, name of producer, city and country.
Letters are too small on the figure 2. Correct it.
In the subsection 3.2. highlight which compounds were responsible for unwanted aromas.
Letters are too small on the figure 3. Correct it.
Letters are too small on the figure 4. Correct it.
Highlight in the subsection 3.3. is it possible by application of different bacterial cultures during milk fermentation to obtain specific profile of aromatic compounds which is the most acceptable by consumers.
Author Response
It is attached here.

Reviewer 3 Report
Comments and Suggestions for Authors
Comments for authors
The paper entitled “Study of Aroma Characteristics and Establishment of Flavor Molecular Labels in Fermented Milks from Different Fermentation Strains” is very interesting research; however, it is necessary to adjust throughout the text:
- What was the reason for labeling the strains with the letters L and P, can you use a more logical nomenclature? not necessarily G and Q, but just one way to differentiate?
- Line 99. Punctuation error
- It should add bibliographic support to the entire materials and methods section.
- Describe the type of sensory evaluation test supporting the use of only panelists (tests).
- You need to better describe the results of figure 1 and 2. The descriptions are not understood and even fig 2 and 3 cannot be read.
- Table No. 3 has an excess of information. Is it possible to summarize only the most relevant information?
- The conclusions are a summary of results, the results obtained are very relevant, you should conclude with the most relevant findings without summarizing the results, align this section with the title and objective of the work. Here you can show the relevance and pertinence of the work.
Author Response
It is attached here.

Reviewer 4 Report
Comments and Suggestions for Authors
The introduction is missing several aspects:
production and consumption statistics for these types of beverages,
L36 Explain how they influence sensory perception?
L46-47 What are these aromas? Mention them.
L51-52 What are these different aromas and how do they influence? I think cognitive aspects such as emotions and memories, which are a key element, are missing.
L56-59 Advantages, disadvantages, and how they complement the sensory evaluation of these analytical techniques should be mentioned.
What is the research hypothesis?
L93-117 Several important elements are missing: 1) What were the standards used for the selection of the people? 2) What preliminary tests were used for the selection of the people? 3) How were the sensory attributes selected? 4) Why were the attributes selected based on the literature and not based on what is perceived in the samples? How can it be ensured that the attributes indicated in the literature are actually perceived? I recommend viewing and citing the following works: https://doi.org/10.3168/jds.2017-14213, 10.1111/joss.12479
How is the use of the 5-point scale justified?
What was the experimental design used to deliver the samples?
How many evaluation sessions were conducted?
What was the statistical model used to evaluate the panel's performance and ensure the results? See and cite: https://doi.org/10.3168/jds.2017-14213
A statistical analysis section should be included, including all the techniques used to evaluate the data.
The figures have low resolution and need to be improved.
Table 3 is not well-presented, and the results cannot be clearly observed. It is recommended to display the correlation tables in a different orientation so that the values ​​can be clearly seen.
What are the limitations of the research?
I see that they used multivariate PLS-R statistics, but confidence ellipses should be included to ensure a difference between samples.
Author Response
It is attached here.

Reviewer 5 Report
Comments and Suggestions for Authors
Paper: “Study of Aroma Characteristics and Establishment of Flavor Molecular Labels in Fermented Milks from Different Fermentation Strains”
by Xu et al., Foods
General comments: In the present work, the authors analysed the impact of different microbial strains on the aromatic characteristics of fermented milk and identified the main volatile compounds to guide strain selection. The work as a whole is very confusing, I suggest the authors in general lighten it up. Below are my suggestions.
Major comments:
- In the abstract, highlight the key findings of the work more explicitly.
- Introduction is very detailed and informative. I suggest the authors to frame the work more concisely and to define its objectives clearly without being too lengthy.
- Inserting an experimental plan in graphic form illustrating the organisation of work would certainly help to provide better insight.
- I understand the amount of data, but table 3 is totally incomprehensible. Please reformat it also in the form of a heat map.
- Paragraphs: 3.1, 3.3 and 3.4 are not adequately discussed deepen with relevant references the results obtained.
Minor comments:
- On line 26 “lactobacillus” should be written in italics.
- In section 2.2. The information included from line 83 to line 88 can be captioned as table 1 because otherwise it only causes confusion.
- On line 89 insert the time and temperature information used for sterilisation and also the equipment used with business information. In addition, a methodological reference regarding the application of this production protocol should be inserted.
- On line 99 remove a dot.
- In section 2.5 insert a methodological reference.
- Table 2 is unclear. I suggest to the authors to enter all the data obtained in the form of a spider plot.
- Improve the design of Figure 2, Figure 3, Figure 4 and Figure 5.
- Include a sentence in the conclusion in which the value of the research is made clear.
Author Response
It is attached here.

Round 2
Reviewer 4 Report
Comments and Suggestions for Authors
Thank you for considering and responding to the comments.
Reviewer 5 Report
Comments and Suggestions for Authors
I have no other suggestions, the authors answered to all suggestions.